# Automated Small River Mapping (ASRM) for the Qinghai-Tibet Plateau Based on Sentinel-2 Satellite Imagery and MERIT DEM

**Xiangan Liang** [1], **Wei Mao** [2], **Kang Yang** [2,3,4] and **Luyan Ji** [5,6,*]

1 Ministry of Education Key Laboratory for Earth System Modeling, Department of Earth System Science, Tsinghua University, Beijing 100084, China
2 School of Geography and Ocean Science, Nanjing University, Nanjing 210023, China
3 Jiangsu Provincial Key Laboratory of Geographic Information Science and Technology, Nanjing 210023, China
4 Southern Marine Science and Engineering Guangdong Laboratory (Zhuhai), Zhuhai 519080, China
5 Aerospace Information Research Institute, Chinese Academy of Sciences, Beijing 100094, China
6 Key Laboratory of Technology in Geo-Spatial Information Processing and Application System, Chinese Academy of Sciences, Beijing 100094, China
* Correspondence: jily@mail.ustc.edu.cn

**Abstract:** The dynamic variation in the water surfaces of the river networks within the Qinghai-Tibet Plateau affects the water resource availability for downstream ecosystems and human activities. Small rivers (with a river width less than 30 m) are an important component of this network, but are difficult to map in the Qinghai-Tibet Plateau. Firstly, the width of most rivers is very narrow, at around 20 m, which appears as only one or two pixels in Sentinel-2 images and thus is susceptible to salt-and-pepper noise. Secondly, local mountain shadows, cloud shadows, and snow pixels have spectral characteristics similar to those of rivers, leading to misclassification. Therefore, we propose an automated small river mapping (ASRM) method based on Sentinel-2 imagery to address these two difficulties. A preprocessing procedure was designed to remove the salt-and-pepper noise and enhance the linear characteristic of rivers with specific widths. A flexible digital elevation model (DEM)-based post-processing was then imposed to remove the misclassifications caused by mountain shadows, cloud shadows, and snow pixels. The ASRM results achieved an overall accuracy of 87.5%, outperforming five preexisting remote sensing-derived river network products. The proposed ASRM method has shown great potential for small river mapping in the entire Qinghai-Tibet Plateau.

**Keywords:** small rivers; Sentinel-2; Google Earth Engine; Gabor filtering; HAND index

## 1. Introduction

Terrestrial river networks store and transport large amounts of water and surface materials, and are an important part of the Earth's biochemical cycle [1–3]. They influence global climate change processes and ecosystem nutrient balances [4,5]. Rivers with channel patterns can be clustered into heterogeneous types based on their morphological characteristics and flow behaviors [6], and can be analyzed through automated and semi-automated approaches [7] using various data sources such as historical maps [8]. With the rapid development of earth observation technology, satellite remote sensing images have become an important data source for monitoring the dynamic changes of terrestrial river networks [9–11].

In recent years, many studies used satellite remote sensing imagery to produce remote sensing data for terrestrial river networks. Yamazaki et al. (2015) produced a Global 3 arc-second Water Body Map (G3WBM, 90 m) dataset using long time series, multi-seasonal Landsat imagery. The Global 3 arc-second Water Body Map (G3WBM, 90 m) dataset used long time series of multi-seasonal Landsat imagery to distinguish permanent from seasonal water bodies based on their frequency of occurrence in the images [12]. Gong et al. (2017) produced a 10-m spatial resolution global land cover product based on Sentinel-2 (10 m) imagery from

2017, and a multiseasonal, multiscale (from 30 m × 30 m to 500 m × 500 m) training sample of global land cover types sampled by visual interpretation using a supervised classification approach, named the Finer Resolution Observation and Monitoring-Global Land Cover (FROM-GLC10) [13], which labelled the global surface water distribution extent with high accuracy. Pekel et al. (2016) used nearly three million Landsat images to produce the Global Surface Water (GSW, 30 m) dataset [14], which contains month-by-month global surface water extraction results from the past 37 years and classifies permanent and seasonal water bodies according to their frequency of occurrence, supporting studies on water conservation management and decision-making, biodiversity conservation, and climate change. GSW records the spatial distribution and dynamics of global water bodies from 1984 to 2020 (37 years). Allen and Pavelsky (2018) produced the Global River Widths from Landsat (GRWL, 30 m) dataset containing river centerline and river width attributes using 7376 Landsat images and field measurements from 3963 hydrographic stations [15]. The total global river area estimated based on the GRWL dataset is more than 44% larger than the results of previous studies. The existing river network remote sensing data products are mainly based on Landsat imageries with a spatial resolution of 30 m. As a result, small rivers with a width of 30 m or less are ignored [16,17]. However, compared with large rivers, small rivers have active ecosystems and frequent land-air interactions [18]. Moreover, ignoring small rivers can cause underestimation of the role of terrestrial river networks in geochemical cycles [19]. Therefore, it is important to use high-resolution satellite imagery to monitor the dynamics of the water surfaces of the small rivers [20].

The Qinghai-Tibet Plateau in China, which accounts for 1/4 of its land area, stores a large number of water resources in the form of glaciers and snow, and is the birthplace of China's major rivers, such as the Yangtze, Yellow, Lantsang, and Tarim rivers [21]. It has a dense network of rivers with different river morphologies, and the spatial and temporal distribution of its rivers dynamically affects the downstream ecosystems and human activities [22–24]. Therefore, water monitoring on the Qinghai-Tibet Plateau is crucial for China's water security. However, the Qinghai-Tibet Plateau has steep terrain, narrow river channels, and widespread small rivers. Additionally, shadows from mountains and clouds and snow cover inevitably influence the remote sensing monitoring of river networks [25,26]. Therefore, developing an advanced method for mapping small rivers on the Qinghai-Tibet Plateau is necessary to improve our understanding of the spatial and temporal dynamic characteristics in water surfaces of the river networks on the Qinghai-Tibet Plateau and quantify their impacts on downstream ecosystems.

## 2. Materials and Methods

### 2.1. Study Area

Zagunao Basin (about 2391 km$^2$) is a sub-basin of the Yangtze River Basin within the Tibetan Plateau (Figure 1) and was selected as the study area. The Yangtze River is one of the five major rivers in the Tibetan Plateau outflow area. The terrain of the study area is steep. The river networks are complex, and the rivers are mostly narrow and small, with an average width of 20 m (Figure 1d) as observed from the existing remote sensing-derived product Finer Resolution Observation and Monitoring-Global Land Cover (FROM-GLC10). Additionally, the study area has an extensive snow cover, and the surface water recharge is snowmelt runoff, which is regulated by temperature. The flood season is usually in summer (June–August), while the dry season is usually in winter (December–February) [27].

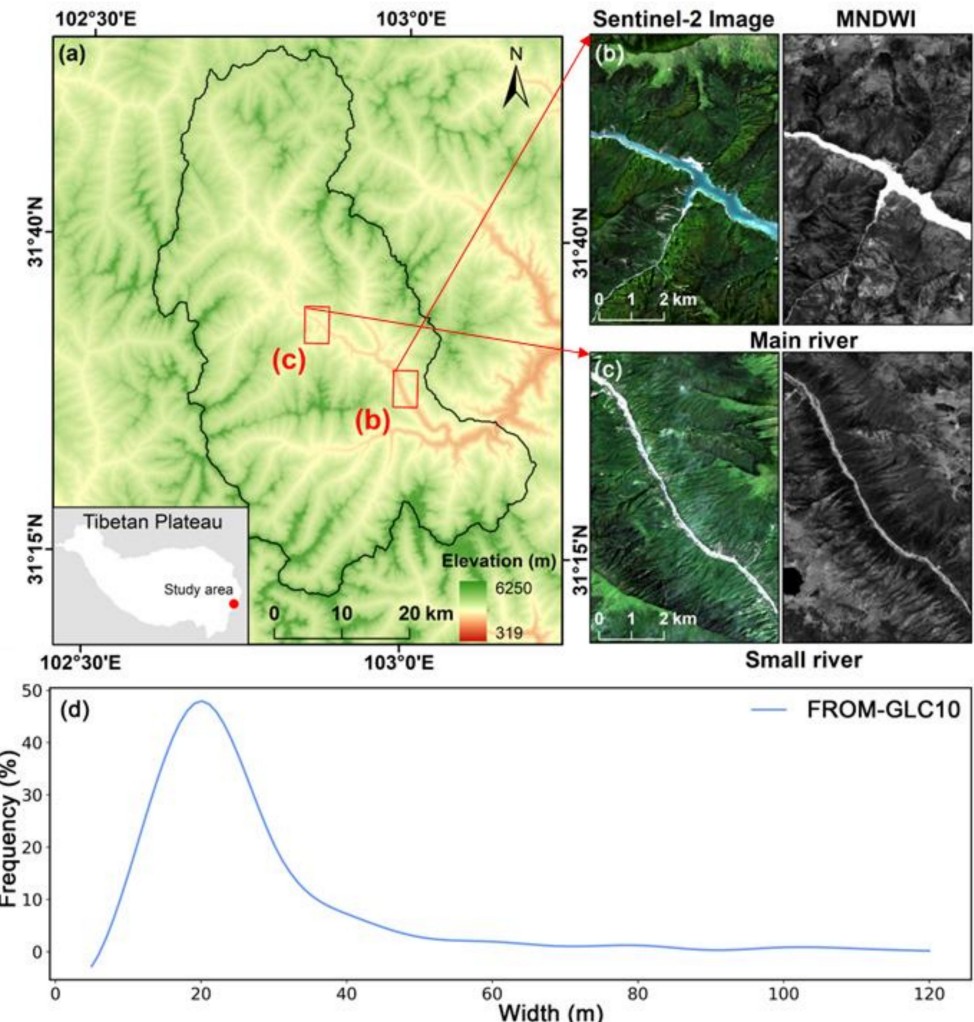

**Figure 1.** Location of the study area. (**a**) Location of the study area; (**b**) Main river channel (the confluence of more than two tributaries with wider river width) within the study area; (**c**) Small river (<30 m) channel within the study area; (**d**) Distribution of river widths in the study area. The river widths were extracted from FROM-GLC10.

### 2.2. Data

#### 2.2.1. Satellite Data

Optical remote sensing images of the Sentinel-2 satellite with 10-m spatial resolution were used for small river monitoring. Sentinel-2 acquires images with a Multispectral Imager (MSI) and is mainly used for terrestrial observation [28]. There are two Sentinel-2 satellites: Sentinel-2A (launched in 2015) and Sentinel-2B (launched in 2017). The Sentinel-2 MSI includes visible and shortwave-infrared bands: Band 2 (blue, 10 m), Band 3 (green, 10 m), Band 4 (red, 10 m), and Band 11 (SWIR, 20 m) [29]. Sentinel-2 has a relatively high spatial resolution among the currently freely available satellite images and has a broad application prospect in the global remote sensing monitoring of environmental resources [30]. Sentinel-2 imagery from May 2017 to October 2020 was acquired and mosaiced to a monthly time step using the GEE platform to produce monthly cloud-free synthetic images for the study area, which were then processed for the extraction and analysis of the river networks.

#### 2.2.2. Digital Elevation Model (DEM)

The MERIT DEM (Multi-Error-Removed Improved-Terrain DEM) is a hydrologically adjusted DEM produced from the MERIT Hydro dataset with a spatial resolution of 30 m

that is particularly suitable for hydrological analysis [31]. We selected MERIT DEM to build the river ROI because it is specially developed for hydrological analysis by reducing error components from existing DEM products, in which river networks and hill-valley structures are clearly represented. The vertical error of MERIT DEM is ±12 m (90th percentile of the error range) [32]. Therefore, we suggest it possesses sufficient accuracy to create river AOIs.

The Height Above the Nearest Drainage Network (HAND) is a topographic index based on topographic data used to simulate the possible occurrence of water bodies [33]. It is widely used in the fields of soil moisture simulation, flood prediction, and assessment [33]. The HAND index has also been used for Arctic river mapping [34]. The higher the difference in elevation, the higher the gravitational potential energy, and the more work is required to transport water from the river to a location, so a lower soil-water content indicates a lower possibility of the presence of a water body [33]. The HAND data produced from the MERIT DEM were obtained from the MERIT Hydro dataset on the GEE platform and used as auxiliary data to remove mountain and cloud shadows as well as snow cover.

The *HAND* index is calculated as follows:

$$HAND = E_i - E_{nearest} \tag{1}$$

where $E_i$ denotes the elevation of each pixel in DEM. The $E_{nearest}$ is the elevation of the nearest pixel of the DEM-modeled river network.

### 2.2.3. Existing Remote Sensing River Network Products

Five existing river network remote sensing products were obtained for comparison with ASRM (Table 1).

**Table 1.** Existing river network remote sensing products.

| Name | Data Source | Resolution | Introduction | Reference |
|---|---|---|---|---|
| FROM-GLC10 | Sentinel-2 MSI | 10 m | Global land cover data products | Gong et al., 2019 [13] |
| GSW | Landsat MSS/TM/ETM+/OLI | 30 m | Long time series monthly surface water cover | Pekel et al., 2016 [14] |
| GRWL | Landsat MSS/TM/ETM+/OLI | 30m | Vector data product with river width | Allen et al., 2018 [15] |
| OSM | Aerial, satellite imagery, GPS devices and in situ observation data | None | Vector data product based on opensource community contributions | https://www.openstreetmap.org/, accessed on 1 January 2021 |
| HydroSHEDS | SRTM DEM | 450, 900 m | Continuous river networks generated by DEM | Lehner et al., 2008 [35] |

(1)  FROM-GLC10: 10-m spatial resolution global land cover product

Finer Resolution Observation and Monitoring-Global Land Cover (FROM-GLC10) is a 10-m spatial resolution global land cover product that includes water bodies identified from Sentinel-2 imagery [13]. The overall accuracy of all land cover types tested was 72.76%.

(2)  GSW: Long-time-series remote sensing dataset

Global Surface Water (GSW) is global surface water remote sensing dataset produced based on nearly three million Landsat images [14]. The dataset contains the global monthly surface water extents for the past 37 years. This dataset contains sub-datasets of monthly surface water extents, the annual frequencies of water bodies, and the maximum water body distribution range, all with a spatial resolution of 30 m. In this paper, the "maximum water body distribution range" data in the GSW dataset were selected for comparison with ASRM.

(3) GRWL: A global river network dataset with river width

Global River Widths from Landsat (GRWL) is a global river network dataset with river width attributes [15]. The dataset is based on over 7000 cloud-free Landsat images acquired during the normal water flow period and field measurements from nearly 4000 hydrological stations, combined with statistical models to model the centerline locations and width of the global rivers. GRWL is the first global river network dataset with river width attributes.

(4) OSM: Real-time update of a global geographic dataset

OpenStreetMap (OSM) is free open-source dataset. Every member of the OSM community can upload and maintain the data, including rivers, lakes, roads, railroads, etc. Contributors combine aerial and satellite imagery, GPS devices, and field work to verify and correct the data to ensure accuracy and timeliness. The river network dataset in vector format from the OSM community was acquired for this paper.

(5) HydroSHEDS: A global river network dataset produced from DEM

HydroSHEDS is a global river network dataset produced using Shuttle Radar Topography Mission (SRTM) DEM data [35]. Since the HydroSHEDS dataset is generated from DEM data modelling, it contains only river centerline information, and lacks information on the actual extents of the rivers.

### 2.2.4. Validation Data

We extracted samples from various months (Figure 2b) and manually classified the pixels as rivers or background (i.e., non-river pixels) referring to the original Sentinel-2 image. A total of 137,357 pixels were selected as validation samples, 73,254 of which corresponded to river pixels.

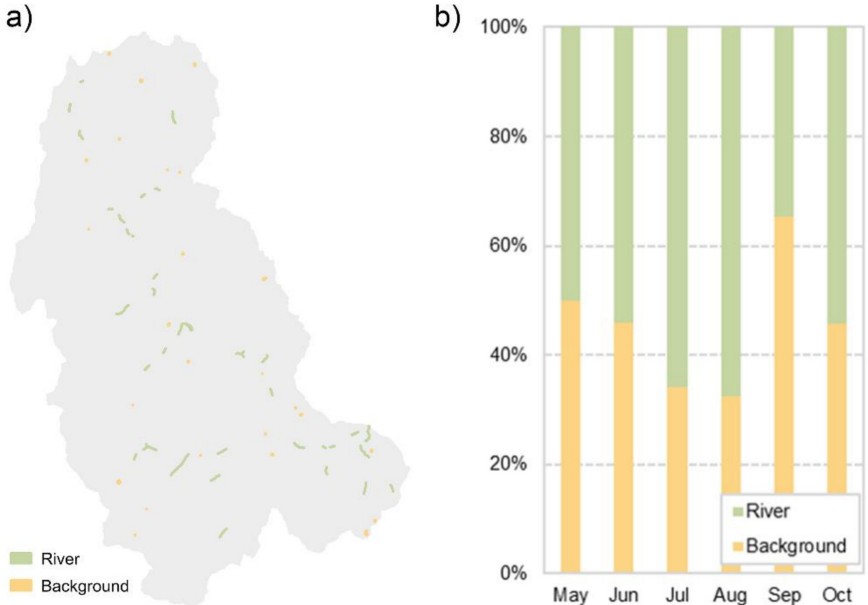

**Figure 2.** Validation samples. (**a**) Distribution of validation samples collected in July 2018; (**b**) Ratio of river pixels to background pixels in samples from various months.

### 2.3. Automated Small River Mapping

Due to the unique climatic and topographical conditions of the Tibetan Plateau, there are two main difficulties in mapping small rivers. Firstly, the widths of the rivers are very narrow (around 20 m) (Figure 1d). They appear in the images as one or two pixels and are susceptible to salt-and-pepper noise from unexpected atmospheric conditions and quality issues with the electronic sensor during the image production process. Secondly, local

mountain shadows, cloud shadows and snow cover have similar spectral characteristics to those of rivers. To address these two problems, a preprocessing procedure was imposed to remove the salt-and-pepper noise and enhance the linear characteristics of the rivers. Additionally, a DEM-based post-processing was imposed to crop the misclassifications caused by mountain shadows, cloud shadows, and snow cover. Then, the river network map generated by ASRM was compared with the existing five products.

### 2.3.1. Water Index

The first step is to calculate the MNDWI (Modified Normalized Difference Water Index) (Figure 3b), which is an improved normalized water index based on the NDWI (Normalized Difference Water Index) [36], both of which take advantage of the spectral characteristics of water bodies with very low reflectance in the short-wave infrared band and high reflectance in the visible bands. The MNDWI is constructed by adjusting the combination of the bands that comprise the NDWI [37]. In particular, MNDWI can better distinguish between shadows and water bodies, and is calculated as follows:

$$\text{MNDWI} = \frac{\text{Green} - \text{SWIR}}{\text{Green} + \text{SWIR}} \qquad (2)$$

where Green and SWIR denote the reflectance in the green and shortwave infrared bands, corresponding to Band 3 and Band 11 of Sentinel-2 images, respectively.

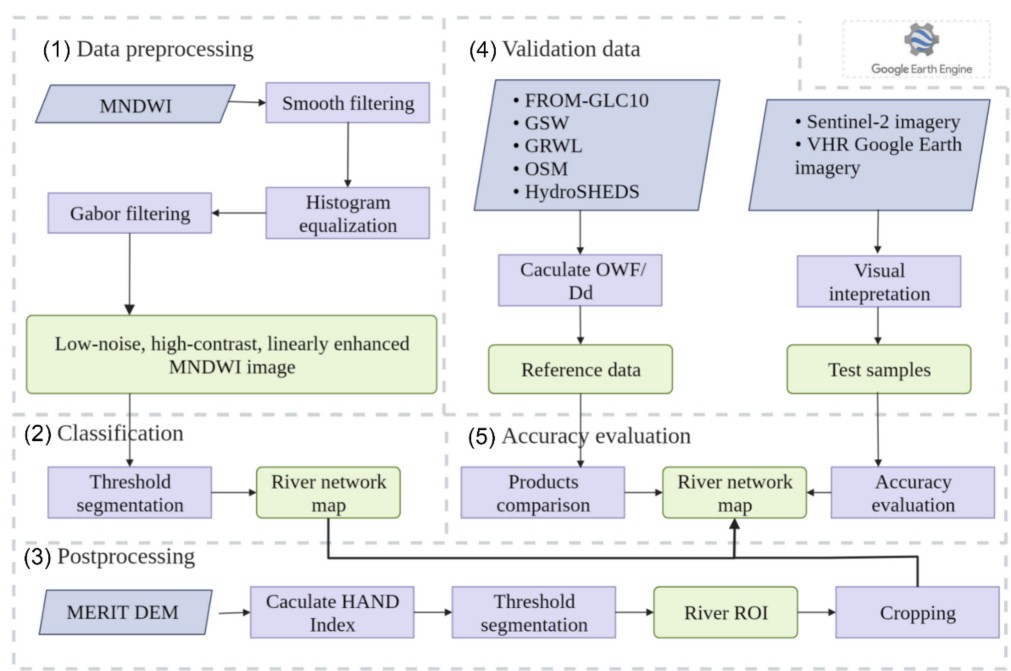

**Figure 3.** Workflow of remote sensing river extraction.

### 2.3.2. River Enhancement

The salt-and-pepper effect increases the difficulty of distinguishing small rivers. Therefore, mean smooth filtering with a window size of $3 \times 3$ is imposed to suppress the salt-and-pepper noise (Figure 4). However, the smoothing process also blurs river boundaries. Therefore, we use a histogram equalization algorithm to enhance the local contrast between the water boundary and the background. Then, a Gabor filter is applied to enhance the linear characteristics.

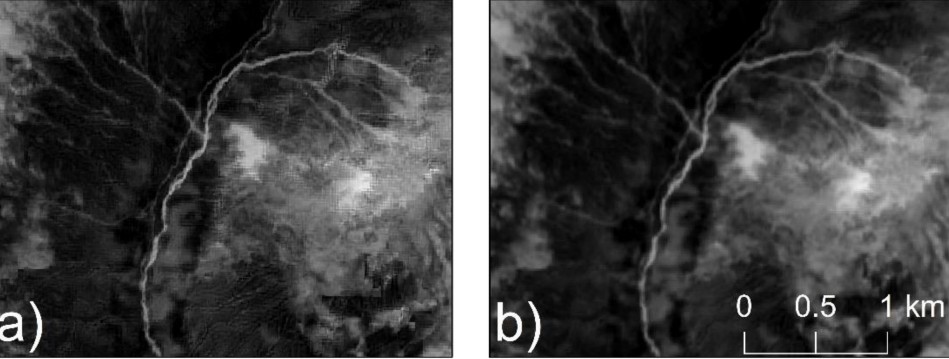

**Figure 4. Salt-and-pepper** noise removal using smooth filtering. (**a**) MNDWI before smoothing. (**b**) MNDWI after smoothing.

A Gabor filter is a band-pass filter that enhances the linear characteristics of a specific width (20 m for this study) while suppressing irrelevant linear elements (e.g., feature edges). It is often used in the delineation of linear blood vessels in medical imaging [38]. Since rivers and blood vessels have similar linear features, it is also used for river identification [39]. Here, Gabor filtering is imposed to enhance the linear characteristics of small rivers in MNDWI images so that the contrast between small linear rivers and background is enhanced.

The Gabor filter kernel with an angle of $\theta = -\pi/2$ can be expressed as:

$$g(x,y) = \frac{1}{2\pi\sigma_x\sigma_y} \exp\left[-\frac{1}{2}\left(\frac{x^2}{\sigma_x^2} + \frac{y^2}{\sigma_y^2}\right)\right] \cos(2\pi f_0 x) \tag{3}$$

where $\sigma_x$, $\sigma_y$ is the standard deviation in the $x$, $y$-direction, and $f_0$ is the frequency of the modulated sinusoidal curve. The setting of these three parameters can be adjusted to the specific width of the linear features to be enhanced. Assuming that $W$ is the width of the river to be enhanced, the bandwidth $w$ of the Gabor filter and $W$ satisfy Equation (3). The values of $\sigma_x$, $\sigma_y$, and $f_0$ satisfy Equations (4) and (5).

$$W = 2w + 1. \tag{4}$$

$$\sigma_x = \sigma_y = \frac{w}{2\sqrt{2ln2}} \tag{5}$$

$$f_0 = \frac{1}{w} \tag{6}$$

Since Gabor filtering is performed in the null domain, it is necessary to place $g(x,y)$ according to a certain angle $\theta$ to enhance the river lines in different directions. The rotation, ($\theta \in [-\pi/2, \pi/2]$) equation is as follows:

$$\begin{cases} g^\theta(x',y') = g(x,y) \\ x' = x\cos\theta + y\sin\theta \\ y' = y\cos\theta - x\sin\theta \end{cases} \tag{7}$$

Depending on the rotation angle, the filter can observe the river in the direction that is not used. For the input image $f(x,y)$, the filter will respond with the following equation:

$$r^\theta(x,y) = g^\theta(x,y) * f(x,y) \tag{8}$$

The final maximum response in all directions is expressed as:

$$r(x,y) = \max\left(r^\theta(x,y)\right), \theta \in \left[-\frac{\pi}{2}, \frac{\pi}{2}\right] \tag{9}$$

After experiments, the filter performs most successfully for rivers with a width of 20 m when $w = 2$, $\theta = 15°$. The result after applying the Gabor filter is shown in Figure 5.

The river is more clearly differentiated from the background, and the connectivity of the river is further improved. Moreover, the range of pixel values is set to 0–255.

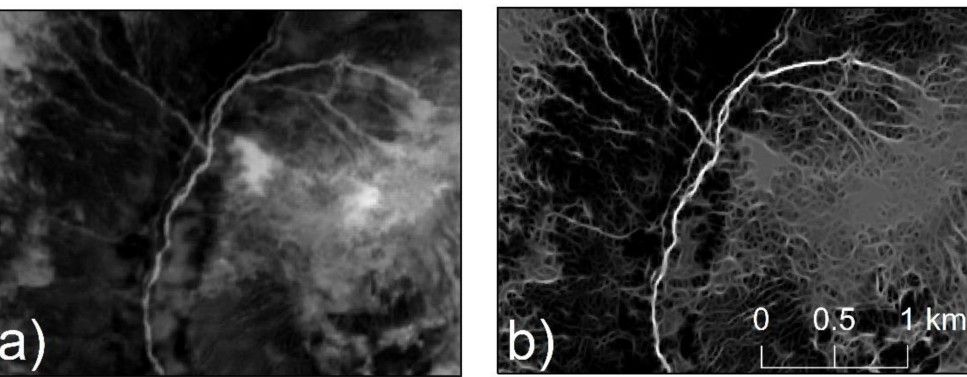

**Figure 5.** Performance of Gabor filtering. (**a**) MNDWI before filtering. (**b**) MNDWI after filtering.

After manual testing at 0.5 intervals, a threshold of 10 was applied on the low-noise high-contrast and on the linear enhanced MNDWI image for river segmentation. Pixels with values higher than 10 were classified as river pixels, while others were classified as non-river pixels. However, the interferences from mountain shadows, cloud shadows, and snow cover remain unremoved and will be handled in the post-processing.

### 2.3.3. Post-Processing

The preliminary remote sensing river networks have been obtained after threshold segmentation of the Gabor-filtered MNDWI (Figure 6b). However, there is still extensive noise caused by mountain shadows, cloud shadows, and snow cover. The spectrum of shadows will vary depending on the underlying surface. The shadow overlying on the vegetation and snow cover would have a quite low reflectance in all bands with a higher reflectance in the visible bands than in the SWIR band, which is an effect similar to that of water. While the shadow overlying on the bare land has a lower reflectance in the visible bands than in the SWIR band, snow has a much higher reflectance in the visible bands than in the SWIR band. Although the reflectance value of snow is greater than that of water (especially the visible part), the MNDWI of snow and water are similar. Since we use the MNDWI index in this paper, both snow and shadows can be misclassified as rivers. Therefore, DEM data were used as auxiliary data to post-process the river extraction results and to remove the noises caused by shadows and snow cover.

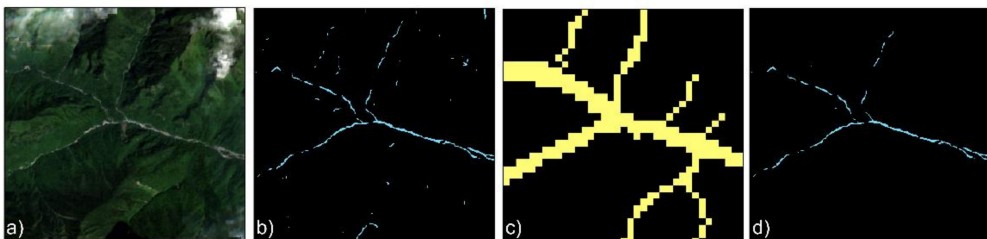

**Figure 6.** Classification removal based on DEM-modelled area of interest (AOI). (**a**) Original Sentinel-2 image. RGB: band 4 (Red), band 3 (Green), band 2 (Blue). (**b**) Results before cropping. (**c**) AOI generated from HAND index. (**d**) Cropped results.

Lu et al. [40] used a DEM to model the central line of a river channel and generated a 50-m buffer on each side of the river. The water pixels outside the AOI were considered to be misclassified. However, the method did not consider different river widths. Creating buffers with uniform distances along different parts of the rivers may cause the partial cropping out of wider tributaries while retaining the noise around the small tributaries.

To overcome this limitation, the HAND index was introduced, which considers the possibility of a river's presence (Figure 6) to create river AOIs of different widths [38]. We used a relatively high threshold of 50 m for the segmentation of HAND imaging to ensure all river pixels are within the AOI, while the pixels with a value of lower than 50 m were considered as a part of river AOI. The AOIs were used to crop the preliminary results of the extraction (Figure 6b) to obtain the final images. After cropping, the noise caused by shadows and snow cover were removed, while the river networks were kept intact (Figure 6d).

### 2.4. Accuracy Evaluation

We constructed a validation sample set by randomly and uniformly sampling a total of 137,357 pixels from different months (Figure 2b), identifying the river pixels by manual visual interpretation and by referring to the original Sentinel-2 image.

Based on the manually interpreted validation data, we calculated the user accuracy, producer accuracy, and overall accuracy to evaluate the reliability of ASRM results. The formulae are as follows:

$$\text{User accuracy} = \frac{Number\ of\ Correct\ river\ pixels}{Number\ of\ Correct + misclassified\ pixels} \tag{10}$$

$$\text{Producer accuracy} = \frac{Number\ of\ Correct\ river\ pixels}{Number\ of\ Total\ river\ pixels} \tag{11}$$

$$\text{Overall accuracy} = \frac{Number\ of\ Correctly\ predicted\ pixels}{Number\ of\ Total\ pixels} \tag{12}$$

The Kappa coefficient is calculated as:

$$\text{Kappa coefficient} = \frac{N \sum_{i=1}^{n} m_{i,i} - \sum_{i=1}^{n}(G_i C_i)}{N^2 - \sum_{i=1}^{N}(G_i C_i)} \tag{13}$$

where $i$ is the number of classes, $N$ is the total number of pixels, $m_{i,i}$ is the number of pixels correctly predicted as class $I$, $C_i$ is the total number of pixels predicted as class $I$, and $G_i$ is the total number of truth values belonging to class $i$.

## 3. Results

### 3.1. Results of ASRM

The overall accuracy of the ASRM reached 87.5%, and the Kappa coefficient was 0.75. The producer accuracy of river classification was 79.25%, and the user accuracy was 96.82% (Table 2), which indicates that there is more leakage of river pixels than misclassification. Most of the unidentified river sections were small rivers with a width less than or equal to the spatial resolution (10 m) of the Sentinel-2 images. These small rivers were represented as mixed pixels in the images, and their spectral characteristics differed less from the background features, rendering them difficult to identify. In addition, we validated the performance per month using the samples collected in the corresponding months (Figure 7).

**Table 2.** Validation of the accuracy of the proposed method (Unit: pixels).

|  | **Background** | **River** | **Overall** | **User Accuracy** |
|---|---|---|---|---|
| Background | 62,194 | 15,197 | 77,391 | 80.36% |
| River | 1909 | 58,057 | 59,966 | 96.82% |
| Overall | 64,103 | 73,254 | 137,357 |  |
| Producer Accuracy | 97.02% | 79.25% |  |  |
| Total accuracy: 87.5% Kappa coefficient: 0.75 |  |  |  |  |

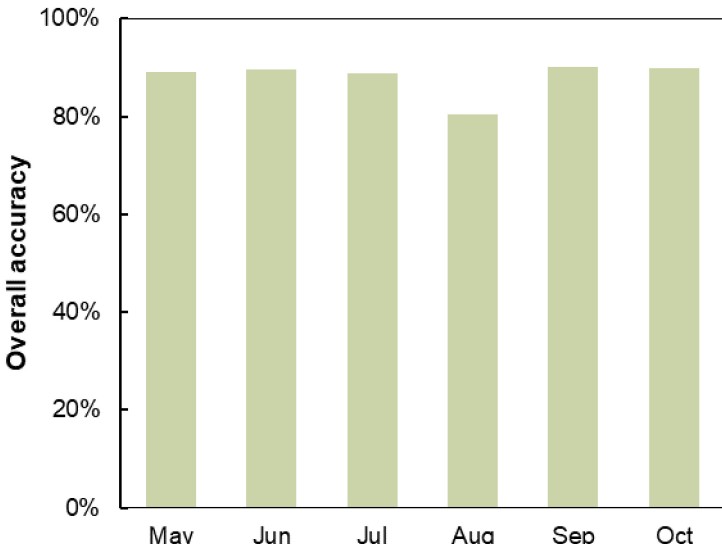

**Figure 7.** Validation of the accuracy of the proposed method per month using the samples collected in the corresponding months.

Further, we evaluated the performance of the ASRM method in different situations. When there were residential areas around the river (Figure 8), the residential areas would potentially affect river mapping, as the edge of the buildings would be enhanced through the Gabor filtering. ASRM identified the intact river channels and removed the residential areas with non-linear characteristics.

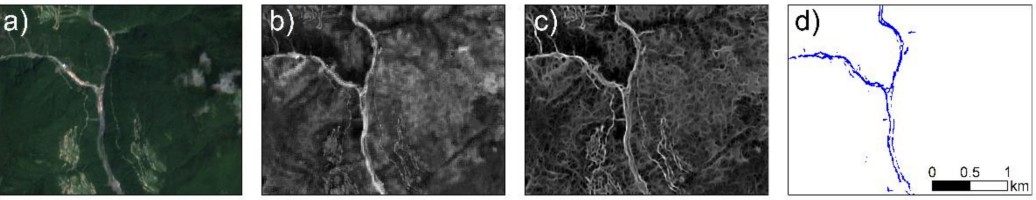

**Figure 8.** ASRM results in the presence of residential areas. (**a**) Original Sentinel-2 image. RGB: band 4 (Red), band 3 (Green), band 2 (Blue). (**b**) MNDWI image. (**c**) Gabor-filtered MNDWI image. (**d**) ASRM results.

To eliminate the influence of mountain shadows (Figure 9), Gabor filtering can enhance the contrast between linear rivers and the nearby pixel clusters of mountain shadows. The post-processing further cropped the mountain shadows far from the central line of the river channel.

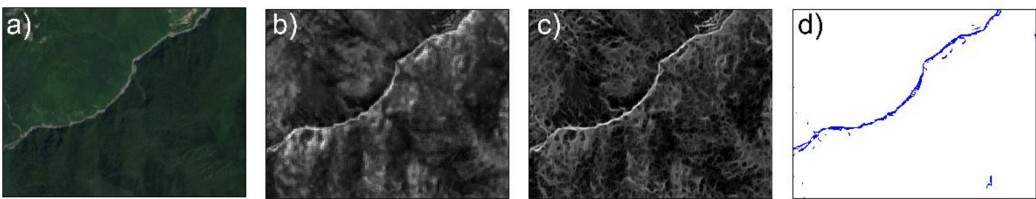

**Figure 9.** ASRM results in the presence of mountain shadow. (**a**) Original Sentinel-2 image. RGB: band 4 (Red), band 3 (Green), band 2 (Blue). (**b**) MNDWI image. (**c**) Gabor filtered MNDWI image. (**d**) ASRM results.

In the presence of cloud cover (Figure 10), although the spectral characteristics of clouds and cloud shadows are similar to those of rivers, the result contained few misclassifications. The misclassifications caused by cloud cover were cropped by the DEM-modelled AOI because they were far from the central line of the river.

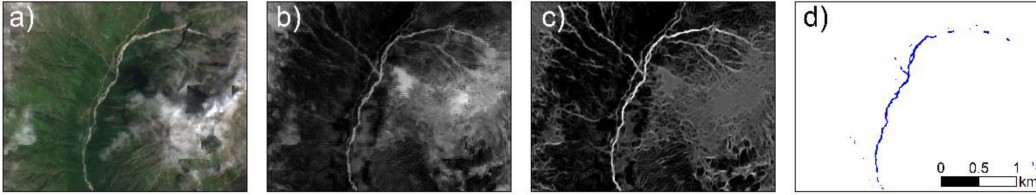

**Figure 10.** ASRM results in the presence of cloud cover. (**a**) Original Sentinel-2 image. RGB: band 4 (Red), band 3 (Green), band 2 (Blue). (**b**) MNDWI image. (**c**) Gabor filtered MNDWI image. (**d**) ASRM results.

### 3.2. Assessment against Existing River Network Products

Five existing remote sensing data products of river network (FROM-GLC10, GSW, GRWL, HydroSHEDS, OSM) were quantitatively compared with ASRM (Figure 11). The results demonstrate that the accuracy of ASRM is superior to those of the other data products (Table 3). In addition, to compare the completeness and continuity of river networks in different data products, drainage density (Dd, the ratio of river length to watershed area) and Open Water Fraction (OWF, the ratio of water body area to watershed area) (Figure 12) were calculated for each product. For the GSW product, we utilized the maximum water extent to form the comparison. For other products with only one period of results, we consider them to represent the maximum extent of water bodies detected in these datasets. We compared them with the ASRM results of 2018.08, which has the highest OWF among results of all months.

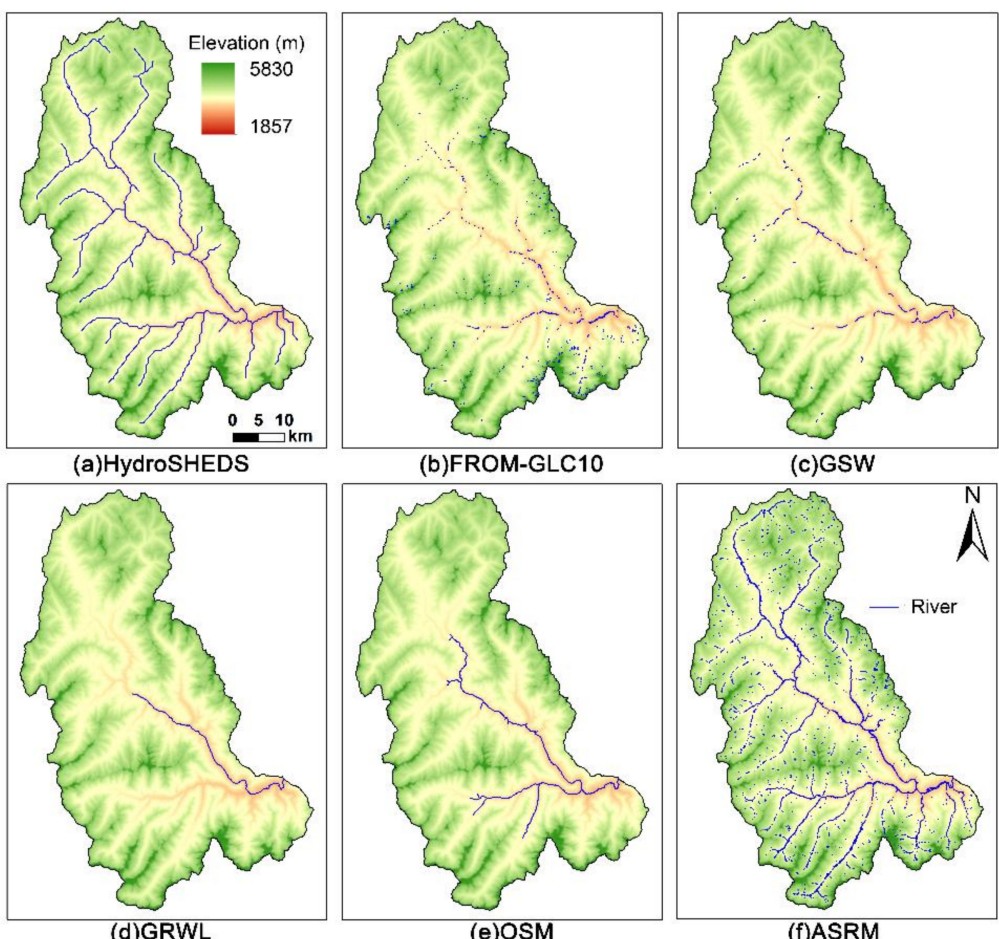

**Figure 11.** Comparison of ASRM with other remote sensing data products.

**Table 3.** Comparison of accuracy of various products.

| Product | Producer Accuracy | User Accuracy | Total Accuracy |
|---|---|---|---|
| ASRM [1] | 79.25% | 96.82% | 87.5% |
| FROM-GLC10 | 4.8% | 92.9% | 59.0% |
| GSW | 37.1% | 93.2% | 69.6% |
| GRWL | 11.6% | 100.0% | 57.0% |

[1] Result of the proposed method.

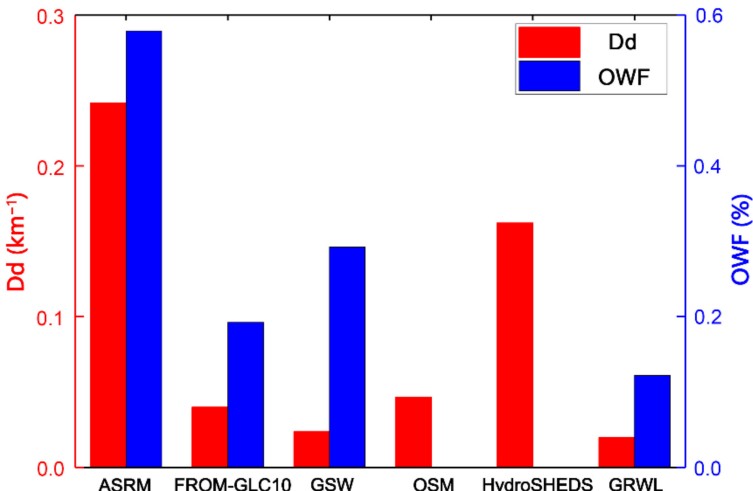

**Figure 12.** Comparison of ASRM with other data products for Dd and OWF.

Compared with FROM-GLC10, GSW, and GRWL, the river networks in ASRM are more complete and more consistent. The Dd of ASRM is 6 times higher than that of FROM-GLC10, 10.17 times higher than that of GSW, and 12.14 times higher than that of GRWL, while only 0.48 times higher than that of HydroSHEDS.

The OWF of ASRM is 6.84 times higher than that of FROM-GLC10, 1.98 times higher than that of GSW, and 12.81 times higher than that of GRWL.

The river width of both ASRM and the FROM-GLC10 dataset reached the peak frequency around 20 m and were mostly distributed in the range of 10–30 m (Figure 13). The peak frequency of river width in the GSW dataset was around 30 m because Landsat images identify rivers less than 30 m. As for the GRWL dataset, the peak frequency of river width is around 70 m.

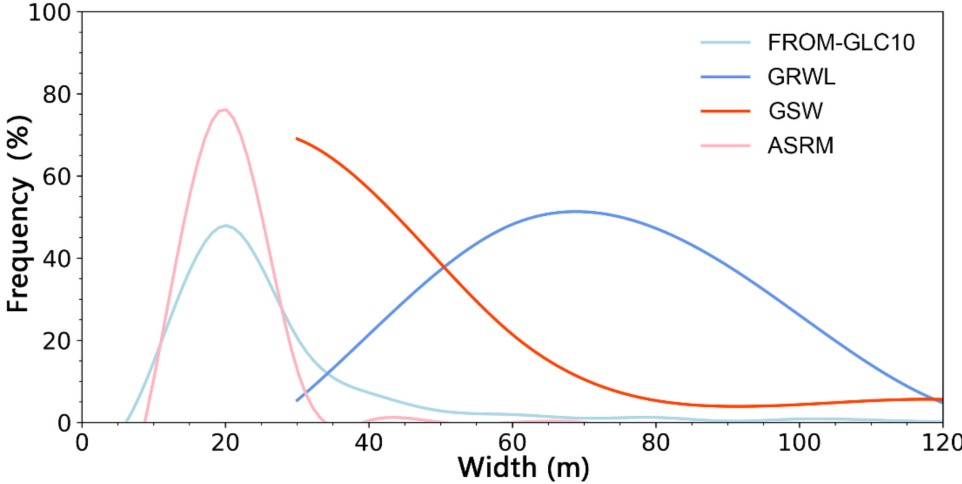

**Figure 13.** Comparison of ASRM with other data products on river width.

## 4. Discussion

### 4.1. Advantages of the Proposed Method over Existing Datasets

Considering the special geographical condition of the Qinghai-Tibet Plateau, there are two important advances in our approach. Firstly, to address the problem of narrow river channels caused by the steep terrain, we used Gabor filtering to enhance the linear features of small rivers. Secondly, we introduced the HAND index to build AOIs and mitigate the effects of mountain and cloud shadows and snow cover.

Gabor filtering enhanced the linear characteristics of small and medium-sized rivers in the image. Additionally, Sentinel-2 images (10 m) have higher spatial resolution compared with Landsat images (30 m), based on which the GSW and GRWL were produced. Therefore, the Dd of ASRM is much higher than that of GSW, GRWL and FROM-GLC10 (Figure 12). HydroSHEDS is produced from a DEM, which continuity ensures that its rivers features show fewer interruptions than remote sensing results, and for this reason, it should have a higher Dd. However, because it lacks the details of small tributaries, its Dd is still smaller than that of ASRM.

The OWF difference between ASRM and the GSW dataset is relatively small (1.98 times) compared to the Dd difference (Figure 12). The low spatial resolution of the Landsat imagery results in the overestimation of OWF due to the identification of rivers less than 30 m wide as 30 m, and the mixed pixels of river boundaries as rivers.

The study compared the river width attributes of the Sentinel-2 extraction results and other five global river network remote sensing data products (Figure 11) and found that Sentinel-2 could extract more rivers with river widths of less than 30 m, which compensated for the shortcoming of the Landsat imagery, which could only extract rivers wider than 30 m. For both the ASRM and the FROM-GLC10 extraction results, the peak frequency was around 20 m, while the peak frequency of GSW was around 30 m as a result of the Landsat images pixel 30-m resolution, assigning all rivers less than 30 m to a 30-m width. The peak frequency of the river width of GRWL was around 70 m, which is likely because the GRWL data product only identified the main streams with the largest flow connected to the pour point in the study area, while the small tributaries with smaller river widths upstream were not identified.

However, there are still several limitations of this method. First, the producer accuracy of river classification was 79.25% (Table 2), which indicates that a large portion of river pixels were not identified. Compared to the Sentinel-2 image, it is found that most of the unidentified river sections were small rivers with a width less than or equal to the spatial resolution (10 m) of the Sentinel-2 images. These small rivers were represented as mixed pixels in the images, and their spectral characteristics differed less from the background features, rendering them difficult to identify. Second, this method is based on a pixel-wise algorithm, leading to the disconnection in the river networks. To create a real river network, those river pixels must be interconnected to form a continuity. Third, our method can only monitor dynamic variation in the water surface of the rivers, instead of the transformation of the river courses. Therefore, our analysis is limited to the variation of the water surface rather than the geomorphological characteristics of the river network.

### 4.2. Dynamic Monitoring in Water Surface of the River Networks in the Study Area

The main source of surface water recharge in the basin is snowmelt runoff, which is seasonal. The proposed method automatically extracted the study area's river network from May to October during the 2017–2020 period (20 extractions in total, as there were no high-quality images in May, June, and July 2017, and October 2018) (Figure 14). Furthermore, the spatial distribution characteristics, as well as the seasonal and annual dynamic change characteristics of the variation in water surface of the river networks in the study area, were analyzed based on the extraction results.

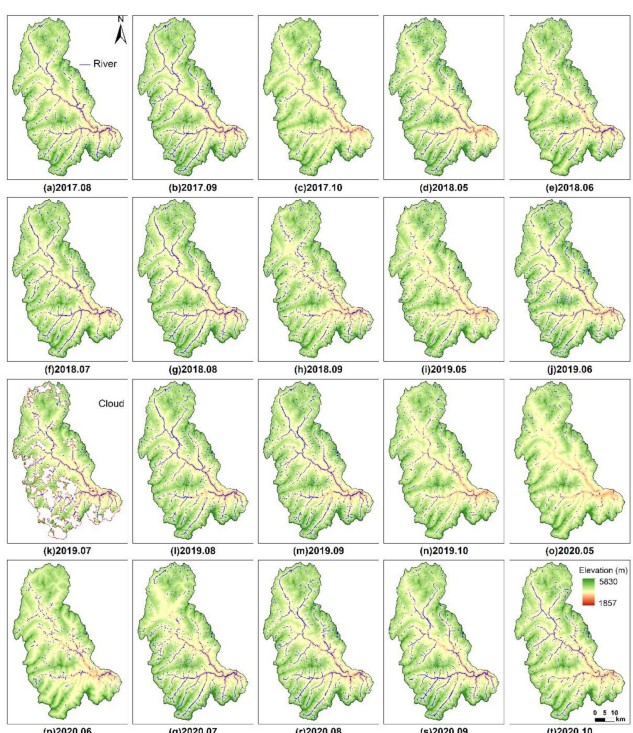

**Figure 14.** Remote sensing river network of the study area, 2017–2020. The cloud area of 2019.07 was cropped, and the remaining area was used to calculate the Dd and OWF.

### 4.2.1. Spatial Distribution Characteristics of the River Networks

The highest elevation along the river network is about 4800 m. The farthest distance from the pour point is 66.4 km (Figure 15). The study areas have steep terrain, and a large proportion (>15%) of the river networks are in areas with slopes greater than 30°. The tributary streams in the upper reaches are narrow and merge into the wider mainstream in the lower reaches. The mainstream with the highest flow and best continuity is distributed near the downstream pour point in the southeast part of the study area. All the water in the basin runs off through the pour point.

### 4.2.2. Dynamic Variation in Water Surface of the River Networks

Since our method can only monitor dynamic variation in water surface of the rivers instead of the transformation of the river courses, our analysis is limited to the variation of the water surface rather than the geomorphological characteristics of the river network.

(1)   Annual Dynamic Variation

From 2017 to 2020, the Dd and OWF of the river networks were stable, exhibiting small annual differences (Table 4, Figure 16). The annual peak of OWF was highest in 2018 (0.58%) and lowest in 2020 (0.52%). In contrast, the peak of Dd in each year did not differ significantly.

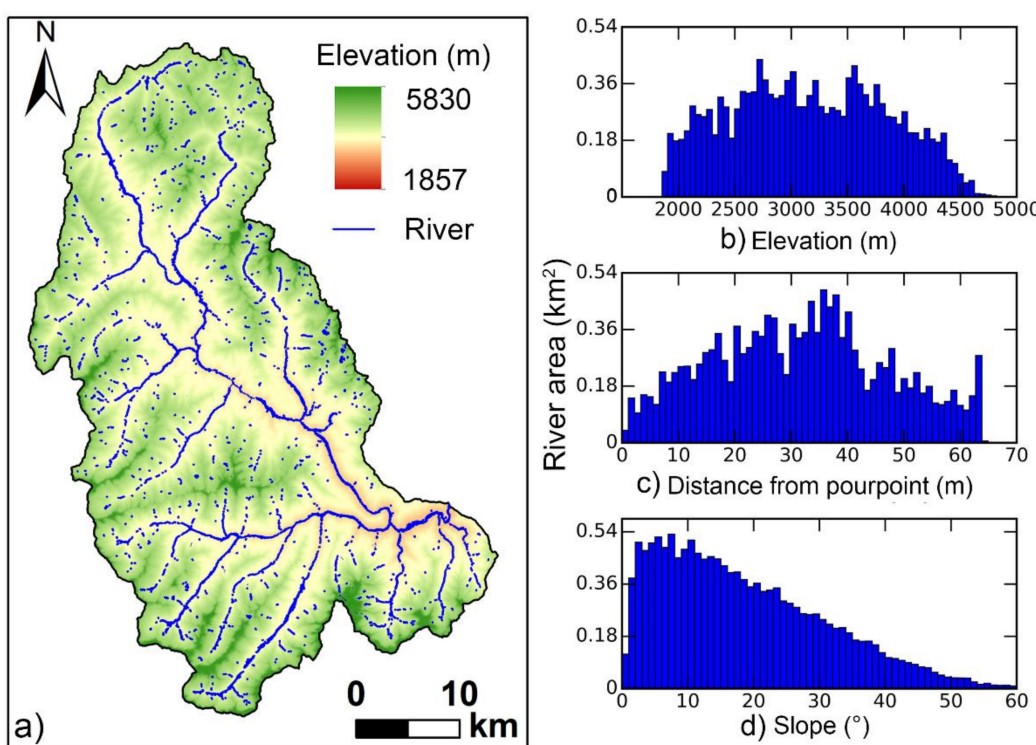

**Figure 15.** Spatial distribution of river networks. (**a**) River mask; (**b**) Elevation of river pixels; (**c**) Distance from water pixels to pour point; (**d**) Slope of river pixels.

**Table 4.** Statistics of OWF, Dd of river networks.

| Date | OWF (%) | Dd (km$^{-1}$) |
|---|---|---|
| 2017/08 | 0.51 | 0.21 |
| 2017/09 | 0.56 | 0.25 |
| 2017/10 | 0.18 | 0.08 |
| 2018/06 | 0.37 | 0.16 |
| 2018/07 | 0.52 | 0.22 |
| 2018/08 | 0.58 | 0.24 |
| 2018/09 | 0.43 | 0.17 |
| 2019/05 | 0.24 | 0.11 |
| 2019/06 | 0.44 | 0.20 |
| 2019/07 | 0.40 | 0.17 |
| 2019/08 | 0.54 | 0.22 |
| 2019/09 | 0.26 | 0.22 |
| 2019/10 | 0.17 | 0.07 |
| 2020/05 | 0.12 | 0.05 |
| 2020/06 | 0.22 | 0.08 |
| 2020/07 | 0.49 | 0.20 |
| 2020/08 | 0.53 | 0.23 |
| 2020/09 | 0.49 | 0.21 |
| 2020/10 | 0.30 | 0.13 |

The trends of Dd and OWF are consistent, both experiencing an increase followed by a decrease during the summer (May to October), with the flood period in July-August and the dry period in May, September, and October.

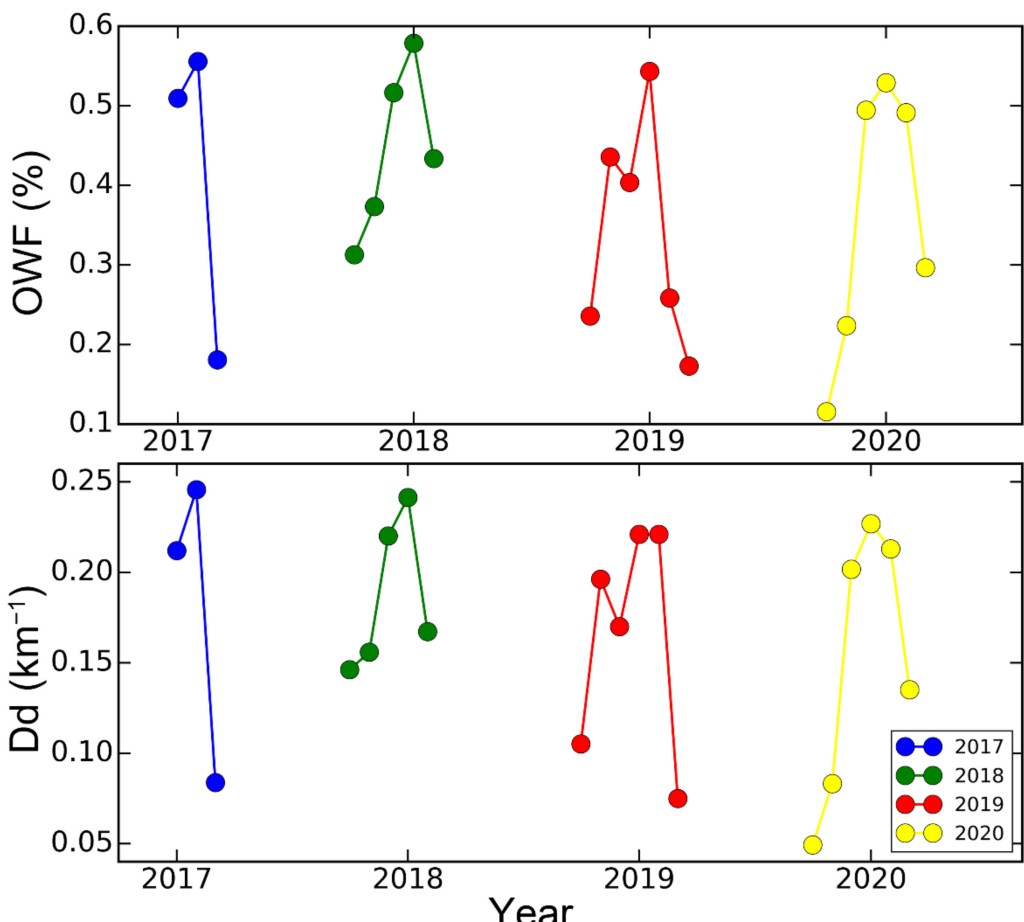

**Figure 16.** Seasonal variation in the water surfaces of the river networks in the study area.

(2)    Seasonal Variation

Based on the proposed method, the dynamic variations in the water surfaces of the river networks in the study area were monitored and well-characterized during each summer between 2017–2020 (Table 5, Figure 17).

**Table 5.** Mean values of OWF and Dd from 2017 to 2020.

| Month | OWF (%) | Dd (km$^{-1}$) |
|---|---|---|
| May | 0.22 | 0.10 |
| June | 0.34 | 0.15 |
| July | 0.47 | 0.20 |
| August | 0.54 | 0.23 |
| September | 0.43 | 0.21 |
| October | 0.22 | 0.10 |

May is the transition period from low to abundant water in the study area, and the lowest Dd and OWF are recorded in May. June is the period of rapid development of water surface, when the OWF is 56% higher than that of May, while Dd is 45% higher. In July and August, the high-water period, both the Dd and OWF reach their peak. September is the transition period from high-water to normal flow; finally, rivers gradually enter the low-flow period in October, and the OWF and Dd decrease by 0.22% and 0.098 km, respectively.

Sentinel-2 imagery displays successful performance in mapping the small rivers within the Qinghai-Tibet Plateau.

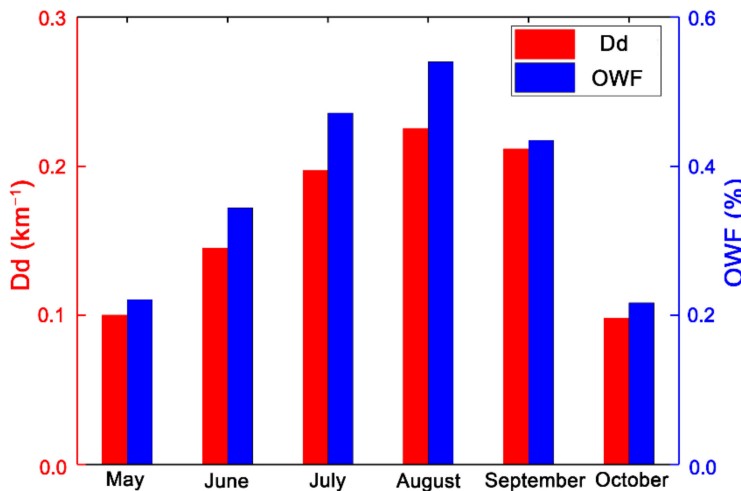

**Figure 17.** Monthly averages of OWF and Dd from 2017 to 2020.

## 5. Conclusions

In this study, we proposed ASRM, an advanced method used for small river mapping using 10-m Sentinel-2 imagery that can overcome the mapping challenges created by the special geographic conditions of the Qinghai-Tibet Plateau. The approach was evaluated with validation samples collected from different months and compared with five existing remote sensing products. Finally, the method was applied to a basin (~2391 km$^2$) within the Qinghai-Tibet Plateau, and the spatial distribution and dynamic variation in the water surfaces of the river networks in the study area from May to October in the period from 2017 to 2020 were analyzed. The results reveal that:

(1) ASRM achieved an overall accuracy of 87.5%, with more leakage of river pixels than misclassifications. The ASRM performed well in the presence of residential areas, mountain shadows, and cloud cover.

(2) Compared to five existing remote sensing products, ASRM identified more small rivers, providing more detailed and consistent maps. The Drainage density (Dd) of ASRM was more than six times that of other datasets, and the Open Water Fraction (OWF) was more than 1.9 times that of other datasets.

(3) From the perspective of interannual variations, the annual variations of maximum Dd and OWF of the river networks in the study area were less than 15% from 2017 to 2020. In terms of the seasonal variations, both Dd and OWF increased from May to August, and decreased monthly after August.

**Author Contributions:** Conceptualization, X.L., L.J. and K.Y.; methodology, X.L. and W.M.; software, W.M.; validation, L.J. and X.L.; formal analysis, X.L. and L.J.; investigation, X.L.; resources, K.Y.; data curation, W.M.; writing—original draft preparation, X.L.; writing—review and editing, L.J.; visualization, X.L.; supervision, L.J. and K.Y.; project administration, L.J.; funding acquisition, L.J. All authors have read and agreed to the published version of the manuscript.

**Funding:** This research was supported by the Second Tibetan Plateau Scientific Expedition and Research Program (STEP), Grant No. 2019QZKK0206.

**Data Availability Statement:** The data presented in this study are available on request from the corresponding author.

**Conflicts of Interest:** The authors declare no conflict of interest.

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
