# Peer review of "Automated Small River Mapping (ASRM) for the Qinghai-Tibet Plateau Based on Sentinel-2 Satellite Imagery and MERIT DEM"

_remotesensing, doi:10.3390/rs14194693_

Round 1

Reviewer 1 Report

Overall, the manuscript is interesting and well-written. The authors provided a new ASRM method that may useful for small river mapping in the entire Qinghai-Tibet Plateau. 

Special comments:

1.  It is better to add two arrows in Figure 1 (a) when zooming in (b) and (c).

2.  I confused with small rivers and main rivers in Line 105, in other words, perhaps you should add more descriptions with them.

3. In line 167, “.....with different seasons....",but in figure 2 the samples are from different months, which is correct?

4. In Figure 12(k), compared with others, why there are so much more white parts in it, could you explain it?

5. In Line 438, "....were stable from 2017 to 2020..." I suggest that it is better to show the data details in section of Conclusions. 

6. Lines 440-441, I recommend to delete the sentences which are not results.

Reviewer 2 Report

The article is methodical. I believe that it should be linked with geomorphological research, where this method can be successfully used.

In the title, do not use the wording `small river`. You do not define a small river in a geomorphological correct way, so it is better to use the term `river with an average width of 20m` (or similar).

In the introduction, after the first paragraph, refer to traditional methods of analyzing river networks and channel patterns. Use: https://doi.org/10.1016/j.geomorph.2020.107558   https://doi.org/10.3390/rs13245147   https://doi.org/10.1016/B978-0-12-374739-6.00263-3

In the `Study area`, add what are the average and extreme flows of the rivers (with an average width of 20 m).

Figure 1. In part (a), add a north arrow. In parts (b) and (c), put captions (b) and (c) in the upper left corner of the figures as you did in (a) and (d).

Lines 167-169. Explain what the background is.

In subsection (2) Seasonal variation (from line 390) there is no geomorphological comment. Geomorphologists can use the proposed method and you must refer to the analysis of channel pattern changes. Refer to typical geomorphological analyzes and show what your method gives and what it lacks. For example in this excerpt "June is the period of rapid river development, when the OWF is 56% higher than that of May, while Dd is 45% higher. During this month, the river networks become more interconnected, and the narrow tributaries in the upper reaches intersect and merge into the mainstream in the lower reaches. " you write about the functioning of the braided river.

Reviewer 3 Report

General comments:

This paper proposes a feasible method for automatic mapping of small rivers on the Qinghai-Tibet Plateau based on Sentinel-2 remote sensing images. The main technical challenges for mapping small river network are: the salt-and-pepper effect on the resultant river network image due to the high resolution required to extract very narrow rivers; anomalous pixels due to mountain shadow, cloud cover and snow cover. The authors developed two procedures to tackle these two challenges and verified the results against existing river network products. The methodology is clearly presented and the results are well validated. However, it also has some flaws, such as the ambiguous definition of river network, lack of necessary information in some sections, as well as some structure problems in the manuscript, which need to be thoroughly revised.

Major concerns:

1 Only preprocessing and postprocessing steps are presented and the classification step is missed in the manuscript.

2 Section 4.2 gives dynamic changes of river network, which should be a part of Results, rather a part of Discussion. Better to move it to the Results section.

3 It is necessary to include analyses of error sources and limitations of this method in the discussion section.

4 The so-called dynamics of river network actually has nothing to do with the river network. Those dynamics are related to variations in the water surface of the rivers. Some rivers fall dry in some seasons, but we cannot say these river courses have been transformed. The term river network is well defined in hydrology. For this paper, the authors could choose a different term to denote this content. Moreover, as shown in figures such as Figs 10-12, the river networks shown are disconnected, which is not correct for a river network. In the proposed methodology, I only see pixel-wise algorithms, to create a real river network, those river pixels must be interconnected to form a continuity.

Some specific comments:

1 Ln 86: "Zagunao Basin is a tributary of the Yangtze River", please note, a basin is not a tributary of a river.

2 Ln 87-89, the description on changes in streamflow of the Yangtze River has nothing to do with this study area.

3 Section 2.2.2: MERIT DEM has a resolution of 30m, much coarser than the working resolution which is 10m. Why not use a 10m DEM? Will this lead to greater uncertainty in the results? I also wonder about the vertical error of MERIT DEM, since the HAND index derived from MERIT is a very important variable to create buffer areas in the post processing step.

4 Ln 120: please provide the equation of HAND index and describe how you calculate this index and how you apply it in the post-processing step, if I am not missing it.

5 Section 2.2.3: provide necessary parameter settings of the mean smoothing filter, such as window size.

6 A threshold classification was applied to achieve preliminary results. Please provide more specs on this step. The current manuscript completely lacks this step.

7 Ln 236-238: I don’t quite understand why shadow and snow cover have similar spectral characteristics to river pixels (water surface pixels).

8 Section 3.1: Since the proposed ASRM method can produce monthly dynamics of water surface, it would be better to validate the performance per months using the samples collected in the corresponding months, in addition to the validations already performed. In addition, since ASRM works on pixels and identifies the water surface, it is possible to produce river width information. Additional validation to this width information would be beneficial. 

9 Ln 278-280: the scene where there are residential areas near the river is mentioned. Please clarify how residential areas potentially affect river mapping.

10 Figure10: These comparisons are not rigorous. The proposed ASRM gives monthly results, what months do those river network products represent? To make a fair comparison, the times the maps represent should match.

11 Ln 320-321: "12.81 times higher than that of HydroSHEDS" is inconsistent with Figure11, where no value for HydroSHEDS of OWF is shown. Better to indicate the values above the bars in the figures.

12 Ln 417-419, it is mentioned that the proposed method could be potentially used for the mapping small rivers on a global scale. It is not the case. This method has been only tested in a mountainous area and relies on elevational discrimination, so it will be problematic when applied to flat regions with dense tiny rivers.  

Round 2

Reviewer 3 Report

I am basically satisfied with the authors’ responses to my question raised in my previous review. The revised manuscript has terms well defined and is much easier to understand. The authors also provided necessary justifications to the methodology established. I recommend an acceptance after some small revisions.

1 I suggest marking important place names on Figure 1a, which may help the readers find the study areas more easily.

2 Figure 3: I don’t understand why a separate symbol of Google Earth Engine is placed on the upper right shoulder of this figure. It looks like not a part of this workflow.

3 Table 1, I suggest adding one more column to reference those products.

4 I noticed the authors use dynamics in (river) water surface in place of dynamics in river network. That is acceptable. But I also found the old usage is still present in other sections, such as in the Conclusions parts where “dynamic variation of the river networks” is used. Also mentioned in Abstract. I really urge the authors can revise thoroughly the manuscript.
